# TALE: Training-free Cross-domain Image Composition via Adaptive Latent Manipulation and Energy-guided Optimization

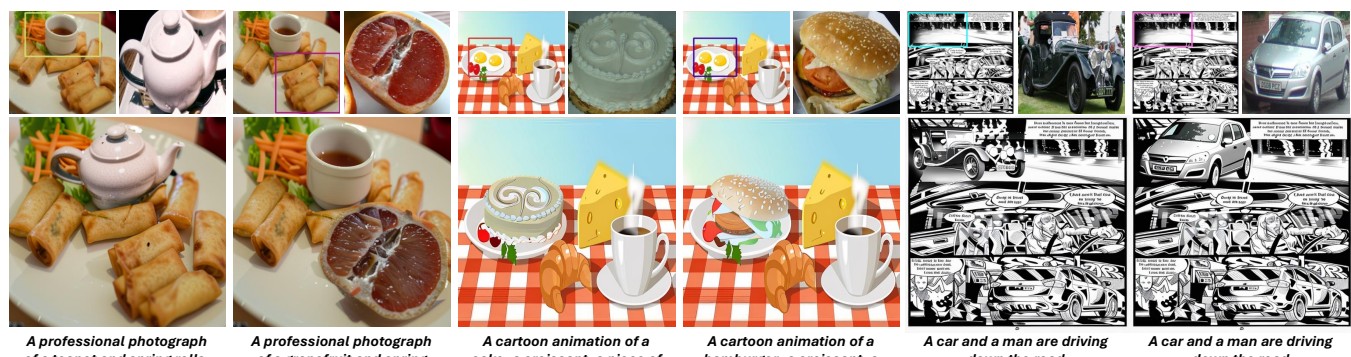

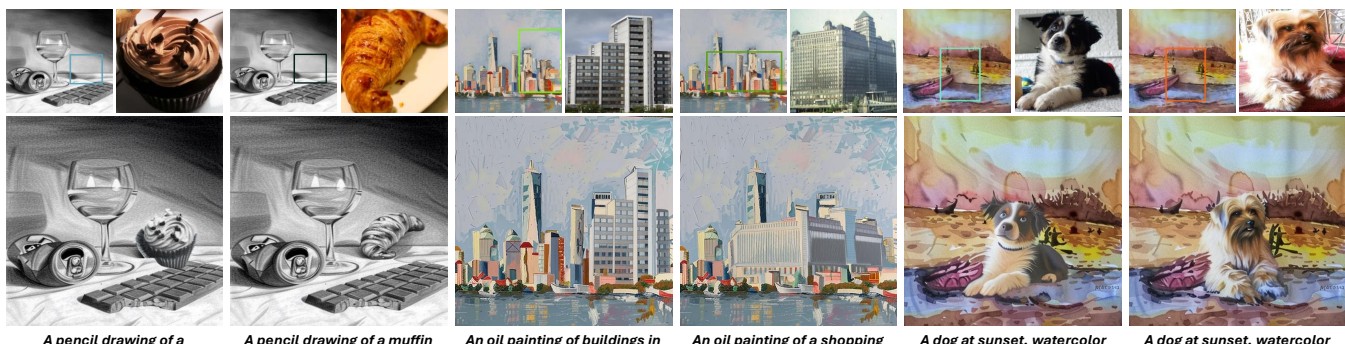

*A professional photograph of a teapot and spring rolls, ultra realistic*

*A professional photograph of a grapefruit and spring rolls, ultra realistic*

*A cartoon animation of a cake, a croissant, a piece of bread and a cup of coffee*

*A cartoon animation of a hamburger, a croissant, a piece of bread and a cup of coffee*

*A car and a man are driving down the road*

*A car and a man are driving down the road*

**(a) Composition within the photorealism domain**

**(b) Composition between real & cartoon domains**

**(c) Composition between real & comic domains**

*A pencil drawing of a croissant and other food, gray tone*

*A pencil drawing of a muffin and other food, gray tone*

*An oil painting of buildings in the distance, Van Gogh Style*

*An oil painting of a shopping mall in the distance, Van Gogh Style*

*A dog at sunset, watercolor painting*

*A dog at sunset, watercolor painting*

**(d) Composition between real & sketching domains**

**(e) Composition between real & oil painting domains**

**(f) Composition between real & watercolor domains**

**Figure 1: Cross-domain image composition targets to harmoniously incorporate objects into specific background context. Our proposed training-free TALE framework enhances text-driven diffusion models with the ability to accomplish this task in diverse domains: (a) photorealism, (b) cartoon animation, (c) comic, (d) sketching, (e) oil painting, and (f) watercolor painting. Zoom-in for more details.**

## ABSTRACT

We present TALE, a novel training-free framework harnessing the power of text-driven diffusion models to tackle cross-domain image composition task that aims at seamlessly incorporating user-provided objects into a specific visual context regardless of domain disparity. Previous methods often involve either training auxiliary networks or finetuning diffusion models on customized datasets, which are expensive and may undermine the robust textual and visual priors of pre-trained diffusion models. Some recent works attempt to break the barrier by proposing training-free workarounds that rely on manipulating attention maps to tame the denoising process implicitly. However, composing via attention maps does not necessarily yield desired compositional outcomes. These approaches could only retain some semantic information and usually fall short in preserving identity characteristics of input objects or exhibit limited background-object style adaptation in generated images. In contrast, TALE is a novel method that operates directly on latent space to provide explicit and effective guidance for the composition process to resolve these problems. Specifically, we equip TALE with two mechanisms dubbed Adaptive Latent Manipulation and Energy-guided Latent Optimization. The former

**Unpublished working draft. Not for distribution.**

formulates noisy latents conducive to initiating and steering the composition process by directly leveraging background and foreground latents at corresponding timesteps, and the latter exploits designated energy functions to further optimize intermediate latents conforming to specific conditions that complement the former to generate desired final results. Our experiments demonstrate that TALE surpasses prior baselines and attains state-of-the-art performance in image-guided composition across various photorealistic and artistic domains.

## CCS CONCEPTS

• **Computing methodologies** → **Image processing**; **Computer vision tasks**; • **Applied computing** → *Arts and humanities*.

## KEYWORDS

Image Composition, Cross-domain, Diffusion Models, Training-free, Adaptive Latent Manipulation, Energy-guided Optimization

## 1 INTRODUCTION

Image composition, as a branch of image editing, has progressively garnered attention in recent years [3, 25, 29, 42, 49, 52]. Typically, this task involves integrating a user-specified image or text prompt into a specified area of background while ensuring that the composited image appears natural and seamless, exhibiting consistent lighting conditions and a smooth foreground-background transition. Image composition has been employed in various fields. For example, the entertainment industry relies on image composition to create stunning visual effects, facilitating the seamless integration of actors and objects into fantastical environments that would be impractical or impossible to capture in real life. Moreover, image composition can also be used in interior design. Specifically, it is used to place virtual furniture into real interior spaces, aiding in visualization and decision-making processes for both designers and clients. In light of these significant and useful applications, it is imperative to explore the field of image composition fully.

The prevailing methods for image composition involve fine-tuning pre-trained models with customized datasets, aiming to improve the semantic coherence of the composited results. For instance, Paint by Example [49] utilizes object detection and data augmentation to generate pairs consisting of a foreground and a background. These pairs are used for training the diffusion model. AnyDoor [3] designs an identity (ID) extractor module to distill the characteristic features of specified objects. These extracted features are subsequently employed as conditional inputs to guide the training process of the diffusion model. While these training-based methods have demonstrated remarkable performance, they require substantial computational effort that limits the accessibility for researchers with constrained resources. Training-free methods (e.g., TF-ICON [29]) offer a promising direction by injection mechanism to merge the self-attention maps of foregrounds and backgrounds. However, they still face critical challenges, particularly in preserving the identity of the composited elements. Besides, they demonstrate subpar performance when tackling cross-domain image composition.

To mitigate these drawbacks, we present TALE, a training-free framework harnessing the power of text-driven diffusion models to tackle cross-domain image composition task, aiming at seamlessly incorporating user-provided objects into a specific visual context regardless of domain disparity. Specifically, TALE functions in the latent spaces, offering precise and potent direction within the compositing workflow to remedy the above-mentioned issues. TALE is equipped with two distinct components: Adaptive Latent Manipulation and Energy-guided Latent Optimization. The former establishes an initial noisy latent conducive to beginning the composition, then applies normalization to iteratively guide subsequent composing steps. In complement, the latter utilizes specific energy functions to further refine the normalized intermediate latents. This synergistic mechanism ensures the production of the intended visual outcomes. The experimental results and user studies reveal that the proposed TALE outperforms existing methods. The code will be made available to promote future research.

Overall, our contributions are listed as follows:

- We propose TALE, a novel training-free framework capable of seamlessly incorporating user-provided objects into diverse visual contexts across multiple domains.
- TALE excels in preserving the identity characteristics of input objects while harmonizing their style with the backgrounds, resulting in highly realistic and aesthetically pleasing composited images thanks to its Adaptive Latent Manipulation and Energy-guided Latent Optimization mechanisms.
- Extensive experiments and user studies provide compelling evidence of TALE's strength over prior work. We will release the code to promote future research.

## 2 RELATED WORK

### 2.1 Image Composition

Image composition is an essential task utilized in various image editing platforms. The primary goal is to integrate an object into a given background [32]. The composition models should create a visually seamless and convincingly realistic composited image, making it imperceptible for observers to discern any traces of manipulation. Generally, image composition can be categorized into two types based on whether the original object's structure or contour is preserved.

When structure preservation is necessary, some works design image harmonization techniques [4, 11, 12, 26, 45, 46], emphasizing color consistency and luminance coherence across the composited areas. Other methods introduce image blending strategies [1, 28, 47, 54] to remedy the unnatural boundaries between the foreground and background, ensuring a seamless integration while maintaining the integrity of the original structure.

Another line of work suggests that maintaining the object's identity is sufficient, allowing for changes in its perspective and enabling more flexibility [3, 21, 24, 25, 29, 42, 43, 49, 52]. For instance, Paint by Example [49] leverages object detection and data augmentation techniques to create foreground-background pairs, with the augmented foreground image acting as a conditioning in the training of a diffusion model. AnyDoor [3] incorporates an ID extractor to capture the identity features of given objects, which are utilized as conditions for training the diffusion model. It is worth mentioning that TF-ICON [29] introduces a training-free framework, taking advantage of pre-trained text-to-image models for

image composition. In particular, it incorporates the self-attention maps extracted when reconstructing foregrounds and backgrounds to melt them together. It is shown that the performance of TF-ICON surpasses existing image composition methods in versatile visual domains, yet they struggle to preserve object identity features and suffer from incohesive style adaptation.

Generally, the proposed TALE adheres to a training-free routine but distinguishes itself from TF-ICON in that our method is capable of well preserving the object identity and seamlessly blending to diverse domains of different styles, powered by the proposed Adaptive Latent Manipulation and Energy-guided Optimization mechanisms.

## 2.2 Diffusion Models

In recent years, diffusion models [2, 10, 13, 14, 33, 35, 38, 40, 53, 57] have become the mainstream of generative models across various domains, owing to their exceptional fidelity and diversity in generated results when compared with GANs [9] and VAEs [19].

Notably, the Latent Diffusion Model (LDM) [37] performs the diffusion process in a VAE-compressed latent space, thereby improving computational efficiency. DDIM [40] introduces a novel approach to accelerate the latent diffusion processes. Remarkably, DDIM inversion has been effectively utilized for editing purposes and has been integrated into other image composition methods [29]. Imagen [38] introduces multiple diffusion models for progressive generation, enhancing the resolution of generated images step by step. SD-XL [35] enlarges the model size and designs curated strategies to enhance the image quality. DiT [33] utilizes Transformers as the backbone and proves the scaling ability. ControlNet [53] introduces an additional branch to receive the additional conditions, such as the canny maps and segmentation maps. Uni-ControlNet [57] enables processing multiple conditions at the same time. Typically, these methods require training or finetuning on the additional conditions involved to enable a certain degree of control over specific tasks.

To enable controllable generation with different conditions at sampling time, several methods leverage energy functions to guide the diffusion process [6, 7, 22, 50, 56], alleviating the cost of training. In particular, EGSDE [56] introduces a time-dependent energy function designated for unpaired image-to-image translation task. Differently, FreeDom [50] proposes a flexible time-independent formulation for energy functions that facilitate different image editing tasks on multiple conditions.

## 3 PRELIMINARY

### 3.1 Latent Diffusion Model

We leverage the pre-trained text-to-image LDM for our composition model. The diffusion procedure follows the standard formulation in [13, 39, 41], which comprises a forward diffusion and a backward denoising process. Given a data sample $\mathbf{x} \sim p(\mathbf{x})$, an autoencoder consisting of an encoder $\mathcal{E}$ and a decoder $\mathcal{D}$ will first project it into latent $\mathbf{z}_0 = \mathcal{E}(\mathbf{x})$. Subsequently, the diffusion and denoising processes are conducted in latent space. Once the denoising is finished and a final clean latent $\hat{\mathbf{z}}_0$ is generated, the sample can then be decoded via $\hat{\mathbf{x}} = \mathcal{D}(\hat{\mathbf{z}}_0)$.

## 3.2 Energy Diffusion Guidance

The original diffusion models [13] can only serve as an unconditional generator. In order to control the generation process with a desired condition $c$, classifier-guided methods [5, 27, 31, 56] propose to alter the prediction of the denoising network as:

$$\epsilon_\theta(\mathbf{z}_t, t, \mathbf{c}) = \epsilon_\theta(\mathbf{z}_t, t) - \sigma_t \nabla_{\mathbf{z}_t} \log p_\phi(\mathbf{c}|\mathbf{z}_t), \tag{1}$$

where $\sigma_t$ is predefined diffusion scalar and $\phi$ is a trained time-dependent noisy classifier that estimates the label distribution of each sample of $\mathbf{z}_t$. The term $\nabla_{\mathbf{z}_t} \log p_\phi(\mathbf{c}|\mathbf{z}_t)$ can be interpreted as a correction gradient that steers $\mathbf{z}_t$ toward a hyperplane in the latent space where all latents are compatible with the given condition $\mathbf{c}$. To approximate such a gradient, a flexible and straightforward way is utilizing the energy guidance function [22, 50, 56] as follows:

$$\nabla_{\mathbf{z}_t} \log p_\phi(\mathbf{c}|\mathbf{z}_t) \propto -\nabla_{\mathbf{z}_t} \xi(\mathbf{z}_t, t, \mathbf{c}). \tag{2}$$

Here $\xi(\mathbf{z}_t, t, \mathbf{c})$ denotes an energy function that quantifies the compatibility between the condition $\mathbf{c}$ and the noisy latent $\mathbf{z}_t$. The more $\mathbf{z}_t$ conforms to $\mathbf{c}$, the smaller the energy value should be. Such a loose property enables great flexibility in designing suitable $\xi$ to suit for each condition $\mathbf{c}$. Correspondingly, the updated conditional backward process can be written as:

$$\hat{\mathbf{z}}_{t-1} = \mathbf{z}_{t-1} - \rho_t \nabla_{\mathbf{z}_t} \xi(\mathbf{z}_t, t, \mathbf{c}), \tag{3}$$

where $\mathbf{z}_{t-1} \sim p_\theta(\mathbf{z}_{t-1}|\mathbf{z}_t)$ and $\rho_t$ is a scale factor. We base on this equation to derive a latent optimization mechanism to modulate the composition process.

## 4 METHOD

### 4.1 Problem Formulation

Given a background (main) image $\mathbf{x}_{bg}$, a foreground (object) image $\mathbf{x}_{fg}$ with segmentation mask $\mathbf{M}_{obj}$, a text prompt $\mathbf{P}$, and a user-provided binary mask $\mathbf{M}_u$ indicating the region of interest within $\mathbf{x}_{bg}$, the objective of cross-domain image composition is to generate composited image $\mathbf{x}_{res}$ that harmoniously acquires three properties. Firstly, the inputted object appears in the masked region of $\mathbf{x}_{res}$ and picks up a similar style to $\mathbf{x}_{bg}$ while preserving its identity features, i.e. $ID(\mathbf{x}_{res} \odot \mathbf{M}_u) \approx ID(\mathbf{x}_{fg})$ and $Style(\mathbf{x}_{res} \odot \mathbf{M}_u) \approx Style(\mathbf{x}_{bg})$. Secondly, the complementing background area of $\mathbf{x}_{res}$ closely resembles the corresponding area of $\mathbf{x}_{bg}$, i.e. $\mathbf{x}_{res} \odot (1 - \mathbf{M}_u) \approx \mathbf{x}_{bg} \odot (1 - \mathbf{M}_u)$. Lastly, the transition area $\mathbf{x}_{res} \odot (\mathbf{M}_u \oplus \mathbf{M}_{obj})$ is visually imperceptible. To concurrently tackle these challenges, we harness the power of pre-trained text-to-image latent diffusion model and propose a novel training-free approach comprised of two stages: Adaptive Latent Manipulation (Section 4.2) to construct and gradually calibrate initial latent suitable for the composition process and Energy-guided Latent Optimization (Section 4.3) to further optimize intermediate latents via task-specific energy function for better outcomes.

### 4.2 Adaptive Latent Manipulation

**Selective Initiation.** To initiate composition process, TF-ICON [29] first inverts $\mathbf{x}_{bg}$ and $\mathbf{x}_{fg}$ into corresponding noisy latent representations $\mathbf{z}_T^{bg}$ and $\mathbf{z}_T^{fg}$ via inversion process of predefined $T$ timesteps. Then, they are merged to constitute noisy latent used as starting

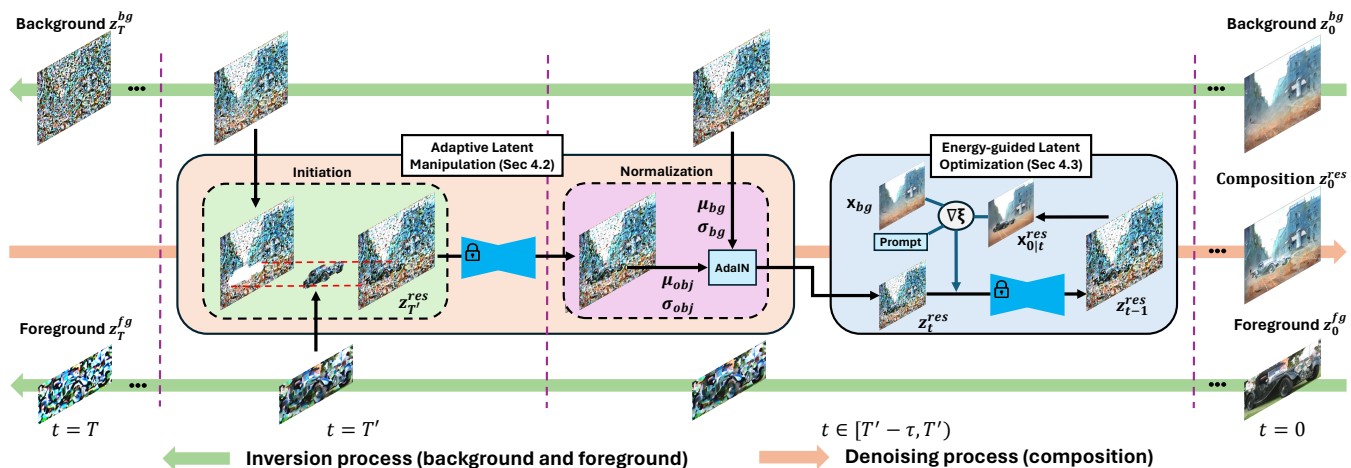

**Figure 2: Illustration for the overall framework of TALE. First, the background latent $z_0^{bg}$ and foreground latent $z_0^{fg}$ are inverted into their respective noisy correspondences $z_T^{bg}$ and $z_T^{fg}$. Then, for selected timestep $T'$, we initiate the composition process by incorporating $z_{T'}^{bg}$ and $z_{T'}^{fg}$ via Selective Initiation (Section 4.2). In subsequent timesteps $t \in [T' - \tau, T']$, the intermediate latent $z_t^{res}$ is progressively refined through the sequential application of Adaptive Latent Normalization (Section 4.2) and Energy-guided Latent Optimization (Section 4.3), ultimately yielding the desired composited result $z_0^{res}$.**

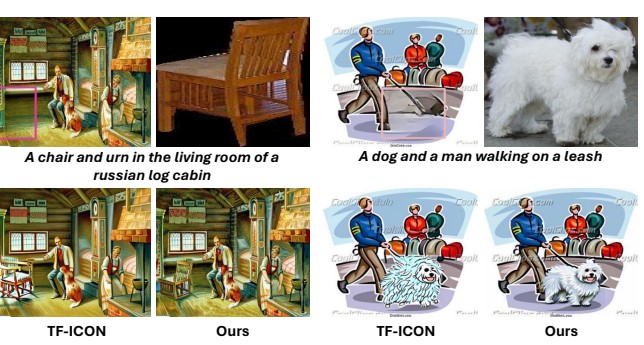

*A chair and urn in the living room of a russian log cabin*

*A dog and a man walking on a leash*

TF-ICON     Ours     TF-ICON     Ours

**Figure 3: Our proposed TALE is robust against identity feature loss and noticeable artifacts indicating domain style disparity compared to TF-ICON.**

point for composing by

$$z_T^{res} = z_T^{bg} \odot M_{bg}^z + z_T^{fg} \odot M_{obj}^z + z \odot M_{tran}^z, \quad (4)$$

where $z \sim \mathcal{N}(0, I)$, $M_{bg}^z = 1 - M_u^z$ indicates region outside $M_u^z$, and $M_{tran}^z = M_u^z \oplus M_{obj}^z$ represents the transition area. Note that these masks are correspondingly rescaled to latent resolution from those mentioned in Section 4.1. After inversion stage, composition is essentially a backward process which involves concurrently denoising $z_T^{bg}$, $z_T^{fg}$, and $z_T^{res}$. The incorporation in Eq. 4 is applied at initial timestep $T$ while for $t < T$, the composition process is implicitly controlled by injecting self-attention maps obtained when denoising $z_t^{bg}$ and $z_t^{fg}$ into those of $z_t^{res}$ in specific manner [29]. Though self-attention maps could bring about some semantic information of the inputted object to the resulting image, they are susceptible to identity features loss and incohesive style adaptation, as illustrated

in Fig. 3. Moreover, randomly initializing values within transition area $M_{tran}^z$ can produce unwanted artifacts.

To overcome these issues, we aim to induce explicit guidance that directly leverages noisy latents to capture identity features better while seamlessly altering domain style. Our empirical observations reveal that it can be achieved by initiating the composition process at a later timestep instead of $T$. Formally, we select timestep $0 < T' < T$ and employ $z_{T'}^{res}$ as the starting point for composing:

$$z_{T'}^{res} = z_{T'}^{bg} \odot (1 - M_{obj}^z) + z_{T'}^{fg} \odot M_{obj}^z. \quad (5)$$

The rationale behind preference of $T'$ over $T$ is the more the denoising progresses, the more style and identity information are reconstructed in $z_{fg}^{T'}$ and $z_{bg}^{T'}$ comparing to those from timestep $T$, hence the more informative and effective they can be brought to $z_{res}^{T'}$. Moreover, the pre-trained denoising network $\epsilon_\theta$ can retain the layout structure of $z_{res}^{T'}$ while gradually rectifying its texture throughout the remaining duration $t \in [0, T')$. Therefore, commencing the composition process at timestep $T'$ with $z_{res}^{T'}$ explicitly leads to desired outcomes without any intervention into self-attention features. Conceivably, this shares a similar intuition with SDEdit [30] to hijack the reverse denoising process, but while SDEdit firstly performs composition in pixel space and then perturbation, we adopt a reversed manner by conducting inversion before composition, allowing for better style harmonization while effectively preserving background and foreground contents. We discuss how to choose the appropriate value for $T'$ in Section. 5.3.

**Adaptive Latent Normalization.** For challenging cases where a significant domain discrepancy exists between $x_{bg}$ and $x_{fg}$, although the Selective Initiation operation is able to integrate identity information of the input object into the composited image, its color hue falls short of the anticipated outcome. For instance, when $x_{bg}$ is black-and-white but $x_{fg}$ is colorful, as in Fig. 6, some colors are

---

**Algorithm 1** Adaptive Latent Normalization

**Input:** Intermediate composited and background latents $(\mathbf{z}_t^{res}, \mathbf{z}_t^{bg})$, preprocessed object segmentation mask $\mathbf{M}_{obj}^z$, threshold $\lambda_t$.

**Output:** Normalized latent $\tilde{\mathbf{z}}_t^{res}$

---

1: $\mu_{bg}, \sigma_{bg} = \mathbf{STATS}(\mathbf{z}_t^{bg})$
2: $\mu_{obj}, \sigma_{obj} = \mathbf{STATS}(\mathbf{z}_t^{res} \odot \mathbf{M}_{obj}^z)$
3: $\mathbf{z}_t^{adn} = \sigma_{bg}(\mathbf{z}_t^{res} \odot \mathbf{M}_{obj}^z - \mu_{obj})/\sigma_{obj} + \mu_{bg}$
4: $\tilde{\mathbf{z}}_t^{res} = \lambda_t \mathbf{z}_t^{adn} + (1 - \lambda_t)(\mathbf{z}_t^{res} \odot \mathbf{M}_{obj}^z) + \mathbf{z}_t^{res} \odot (1 - \mathbf{M}_{obj}^z)$
5: **return** $\tilde{\mathbf{z}}_t^{res}$

---

smeared onto the resulting image. Based on the principle underlying AdaIN [15], we contemplate that tone information is intricately correlated with channel statistics of intermediate latents. Thus, we propose to extract the object region within $\mathbf{z}_t^{res}$, i.e. $\mathbf{z}_t^{res} \odot \mathbf{M}_{obj}^z$, of following timesteps $t \in [0, T']$ and modulate it with channel statistics of background latent $\mathbf{z}_t^{bg}$ via

$$\mathbf{z}_t^{adn} = \sigma_{bg}(\mathbf{z}_t^{res} \odot \mathbf{M}_{obj}^z - \mu_{obj})/\sigma_{obj} + \mu_{bg}, \qquad (6)$$

where $\mu$ and $\sigma$ denote channel-wise means and standard deviations. Besides, we introduce a threshold $\lambda_t$ to further balance the content-style trade-off of the modulated latent as:

$$\tilde{\mathbf{z}}_t^{adn} = \lambda_t \mathbf{z}_t^{adn} + (1 - \lambda_t)(\mathbf{z}_t^{res} \odot \mathbf{M}_{obj}^z). \qquad (7)$$

Finally, substituting $\tilde{\mathbf{z}}_t^{adn}$ into $\mathbf{z}_t^{res}$ results in the updated $\tilde{\mathbf{z}}_t^{res}$ that can preserve content information of object region while its color tone is gradually aligned better with the background.

## 4.3 Energy-guided Latent Optimization

**Energy Function Design.** Despite capturing object identity features and emulating the style of the background, the resulting $\mathbf{z}_t^{res}$ might be inconsistent with the contextual guidance provided by the input text prompt $\mathbf{P}$. This may undermine the rich semantic prior of diffusion model $\epsilon_\theta$ and eventually lead to deviation from intended outcomes similar to TF-ICON. Inspired by [34, 48, 50], we propose to leverage the updated conditional denoising process in Eq. 3 and design suitable energy function $\xi$ to further optimize $\mathbf{z}_t^{res}$ conforming with $\mathbf{P}$. Specifically, given latent variable $\mathbf{z}_t^{res}$ at timestep $t \in [0, T']$, we first derive the composited image $\mathbf{x}_{0|t}^{res}$ from $\mathbf{z}_t^{res}$ and predicted noise $\hat{\epsilon}_t$ via

$$\mathbf{x}_{0|t}^{res} = \mathcal{D}(\mathbf{z}_{0|t}^{res}) = \mathcal{D}((\mathbf{z}_t^{res} - \sigma_t \hat{\epsilon}_t)/\alpha_t), \qquad (8)$$

where $\hat{\epsilon}_t = \epsilon_\theta(\mathbf{z}_t^{res}, t)$ and $\mathcal{D}$ is the decoder mapping from latent back to image space. With such clean prediction on image space, we can then employ external models pre-trained on normal data to estimate $\xi(\mathbf{z}_t^{res}, t, \mathbf{P})$ as below:

$$\xi(\mathbf{z}_t^{res}, t, \mathbf{P}) \approx \mathcal{F} = 1 - cos(\mathbf{EMB}_\mathcal{P}(\mathbf{x}_{0|t}^{res}), \mathbf{EMB}_\mathcal{P}(\mathbf{P})). \qquad (9)$$

Here $\mathbf{EMB}_\mathcal{P}$ projects input into an aligned embedding space via pre-trained multimodal projector $\mathcal{P}$, and $\mathcal{F}$ denotes a distance measuring function, which is one minus cosine similarity between two embedding vectors. The obtained distance then serves as a global penalty to backpropagate the computational graph and obtain a

---

**Algorithm 2** Energy-guided Latent Optimization

**Input:** Intermediate composited latent $\mathbf{z}_t^{res}$, background image $\mathbf{x}_{bg}$ and latent $\mathbf{z}_t^{bg}$, user-specified mask $\mathbf{M}_u$, preprocessed object segmentation mask $\mathbf{M}_{obj}^z$, predefined diffusion scalars $(\sigma_t, \alpha_t)$, prompt $\mathbf{P}$, optimization steps $N$, scale factors $(\eta, \eta')$.

**Output:** Optimized latent $\hat{\mathbf{z}}_{t-1}^{res}$

---

1: **for** $i = 0$ to $N$ **do**
2: $\quad \tilde{\mathbf{z}}_t^{res}, \hat{\epsilon}_t = \mathbf{DENOISE}(\mathbf{z}_t^{res})$
3: $\quad \mathbf{x}_{0|t}^{res} = \mathcal{D}((\mathbf{z}_t^{res} - \sigma_t \hat{\epsilon}_t)/\alpha_t)$
4: $\quad \mathcal{F} = 1 - cos(\mathbf{EMB}_\mathcal{P}(\mathbf{x}_{0|t}^{res}), \mathbf{EMB}_\mathcal{P}(\mathbf{P}))$
5: $\quad \mathcal{F}' = ||\mathbf{G}_\mathcal{P}(\mathbf{x}_{0|t}^{res} \odot \mathbf{M}_u) - \mathbf{G}_\mathcal{P}(\mathbf{x}_{bg})||_F^2$
6: $\quad \bar{\mathbf{z}}_t^{res} = \tilde{\mathbf{z}}_t^{res} - (\eta \nabla_{\mathbf{z}_t^{res}} \mathcal{F} + \eta' \nabla_{\mathbf{z}_t^{res}} \mathcal{F}') \odot \mathbf{M}_{obj}^z$
7: **end for**
8: $\hat{\mathbf{z}}_{t-1}^{res} = \bar{\mathbf{z}}_t^{res} \odot \mathbf{M}_{obj}^z + \mathbf{z}_t^{bg} \odot (1 - \mathbf{M}_{obj}^z)$
9: **return** $\hat{\mathbf{z}}_{t-1}^{res}$

---

gradient on $\mathbf{z}_t^{res}$. By incorporating Eq. 3 and Eq. 9, we can derive the updated composition process as:

$$\hat{\mathbf{z}}_{t-1}^{res} = \mathbf{z}_{t-1}^{res} - \eta \nabla_{\mathbf{z}_t^{res}} \mathcal{F}, \qquad (10)$$

in which $\eta$ serves as the learning rate of each optimization step. We leverage CLIP [36] model with powerful text-image alignment capability as the projector.

Note that $\xi$ can be approximated by a combination of multiple distance functions, one can also compute the distance of the style information between $x_{0|t}^{res}$ within $\mathbf{M}_u$ (the object patch) and $x_{bg}$ to attain better local style cohesion:

$$\mathcal{F}' = ||\mathbf{G}_\mathcal{P}(\mathbf{x}_{0|t}^{res} \odot \mathbf{M}_u) - \mathbf{G}_\mathcal{P}(\mathbf{x}_{bg})||_F^2, \qquad (11)$$

where $\mathbf{G}$ denotes the Gram matrix [17] of the feature map obtained from the projector $\mathcal{P}$ that captures the second-order style information. This extra regularization can be added to Eq. 10 as:

$$\hat{\mathbf{z}}_{t-1}^{res} = \mathbf{z}_{t-1}^{res} - \eta \nabla_{\mathbf{z}_t^{res}} \mathcal{F} - \eta' \nabla_{\mathbf{z}_t^{res}} \mathcal{F}'. \qquad (12)$$

Since object area is the region to be edited while background must remain unchanged, it is intuitive to only optimize the object patch within $\mathbf{M}_{obj}^z$ using Eq. 12, while background region outside the mask can be effectively maintained via replacement trick as in [29].

**Timestep Constraint.** It is observed that applying normalization and optimization for every timestep $t \in [0, T']$ may lead to noticeable artifacts in transition area. Thus, similar to [1, 29], we introduce threshold $\tau$ to regulate them within $t \in [T' - \tau, T']$ only, allowing sufficient time left for diffusion model to rectify the outputs.

## 5 EXPERIMENTS

## 5.1 Experimental Setups

**Baseline Benchmark.** We utilize the benchmark dataset provided by the TF-ICON [29] for evaluation of our method. It includes 332 samples, each comprising a background image, an object image, a user-provided mask, an object segmentation mask, and a text prompt. The background images are divided into four visual domains: photorealism, pencil sketching, oil painting, and cartoon

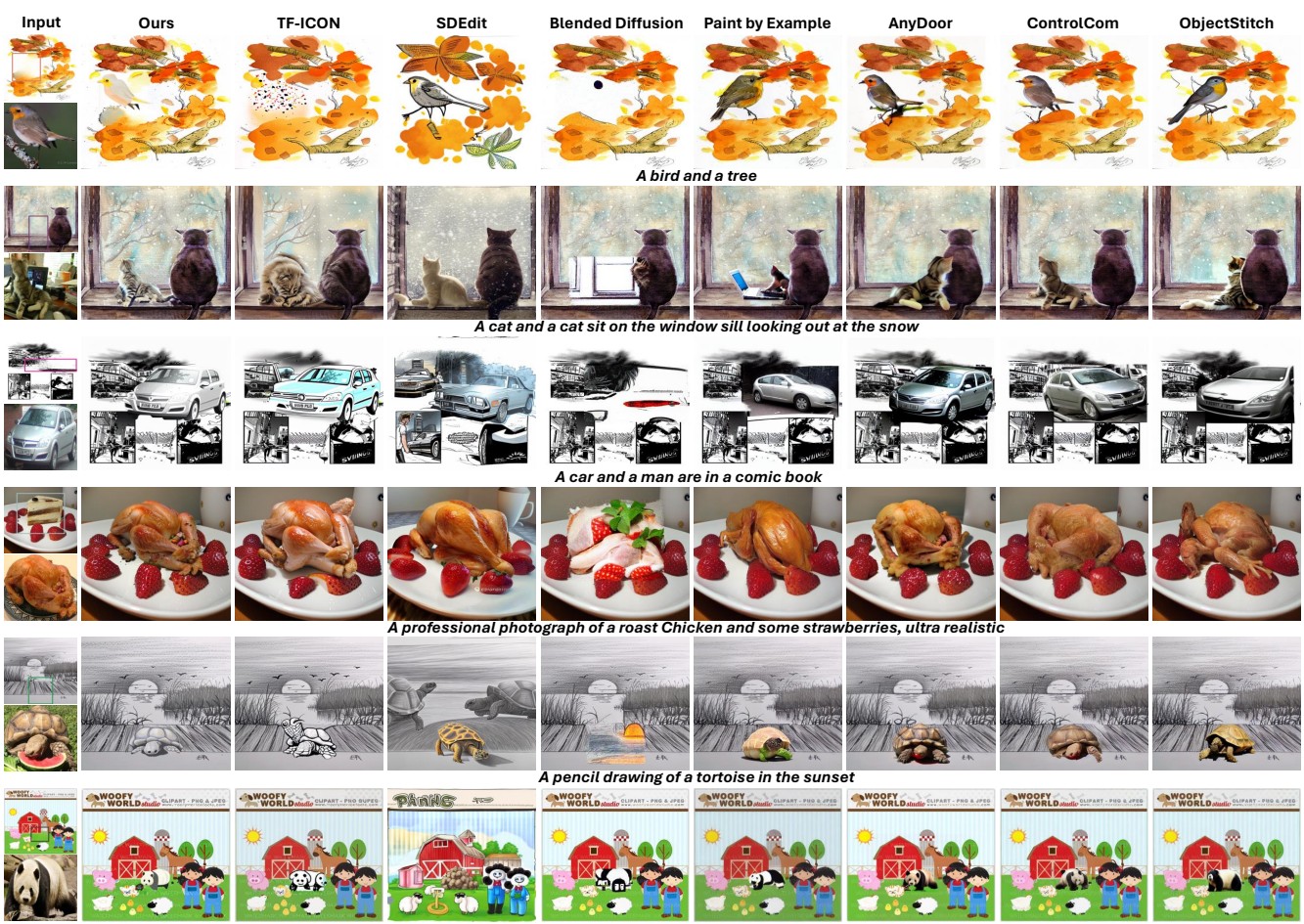

**Figure 4: Qualitative comparison of TALE with prior SOTA and concurrent works in cross-domain image-guided composition. From top to bottom are representative results for compositing between real and watercolor, oil painting, comic, photorealism, sketching, and cartoon animation domains. Zoom-in for details.**

animation. The object images comprise more than 60 categories from photorealism domain with segmentation masks obtained using SAM [20] model. The text prompts are manually annotated according to the semantics of background and object images.

**Extended Dataset.** Since the baseline benchmark is heavily skewed towards the photorealism domain with over 70% of samples and provides a limited number of background images for assessment, we propose an extended dataset with more non-photorealistic samples and diverse backgrounds. We randomly select artistic domain images from Clipart1k, Watercolor2k, and Comic2k [16], to be background images, utilizing their object bounding box annotations for user-specified mask generation. For each background, we randomly select an object of class [CLS] and adopt BLIP2 [23] model to generate caption of template "A [CLS] and …". Then, we leverage Inpaint Anything [51] framework to inpaint the selected object location, obtaining a clean background image. Besides, object images are sampled from the baseline benchmark due to their category diversity. Subsequently, we pair the object and background images,

and accordingly replace [CLS] in the background caption with the category [CLS*] of the paired object. Lastly, we manually remove unreasonable pairs for sanity and eventually obtain an extended benchmark of 207 high-quality non-photorealistic domain samples with diverse backgrounds for evaluation, complementing what is lacking from the baseline.

**Implementation Details.** We first adopt the preprocessing pipeline from TF-ICON [29] to preprocess each data sample so that the input object is rescaled and relocated to correspond with user-inputted mask. In addition, we employ Inpaint Anything [51] model to remove unwanted objects underneath the user mask to produce a clean background image for composing. Then, we conduct composition processes using our proposed training-free approach TALE of which the overall framework is depicted in Fig. 2. Specifically, we leverage the inversion technique introduced in [29] to invert background and foreground images into latent representations $\mathbf{z}_T^{bg}$ and $\mathbf{z}_T^{fg}$ then iteratively denoise them for $T = 20$

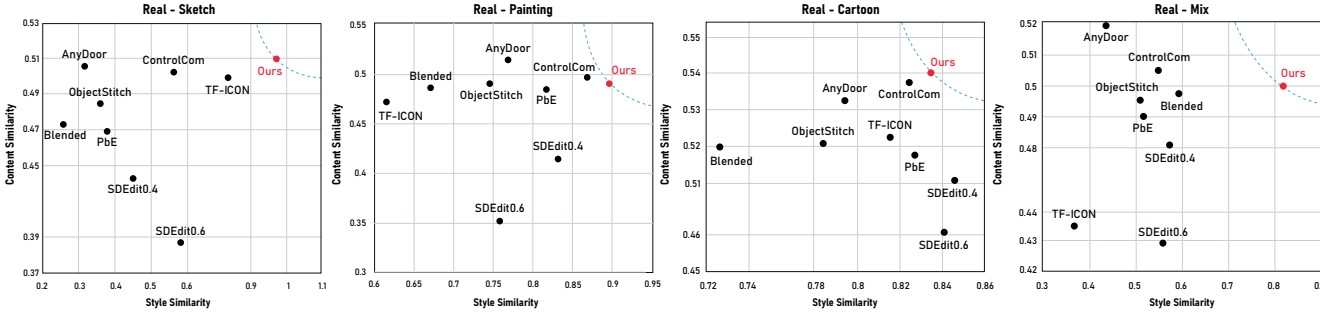

**Figure 5: Quantitative comparison of TALE with prior SOTA works in cross-domain composition on the baseline benchmark with sketching, oil painting, and cartoon animation domains, and on the extended benchmark containing mixture of other domains such as comic and watercolor painting.**

**Table 1: Quantitative performance achieved by different methods for photorealism same-domain composition on test benchmark provided by [29]. Our results are shown in bold, the best and second-best results are in red and blue.**

| | Method | LPIPS$_{bg}$ ↓ | LPIPS$_{fg}$ ↓ | CLIP$_{Image}$ ↑ | CLIP$_{Text}$ ↑ |
|---|---|---|---|---|---|
| **Training** | PbE [49] | 0.12 | 0.69 | 80.26 | 25.92 |
| | AnyDoor [3] | 0.09 | 0.59 | 87.87 | 31.24 |
| | ControlCom [52] | 0.10 | 0.60 | 84.97 | 30.57 |
| | ObjectStitch [42] | 0.11 | 0.66 | 84.86 | 30.73 |
| **Training-free** | Blended [1] | 0.11 | 0.77 | 73.25 | 25.19 |
| | SDEdit (0.4) [30] | 0.35 | 0.62 | 80.56 | 27.73 |
| | SDEdit (0.6) [30] | 0.42 | 0.66 | 77.68 | 27.98 |
| | TF-ICON [29] | 0.10 | 0.60 | 82.86 | 28.11 |
| | **TALE (Ours)** | **0.10** | **0.51** | **85.12** | **31.03** |

timesteps while conducting composition process intertwine starting from $T' = 8$. Subsequently, we proceed to normalize and optimize the intermediate composited latent $z_t^{res}$ via our proposed Adaptive Latent Normalization (Algorithm 1) and Energy-guided Latent Optimization (Algorithm 2) operations with $\tau = 5$. We respectively set $\lambda_t = 0.1 + 0.5(T' - t)/\tau$ for normalization and $N = 3, \eta = 15, \eta' = 0.15$ for optimization. We fix the random seed for fair comparisons and conduct all experiments on NVIDIA Geforce RTX 3090 GPUs, where the composition takes about 23 seconds per sample, depending on the size of the foreground image and user mask. Note that these settings are kept by default for every cross-domain experiments, and for same-domain composition, we adjust $T' = 6, \tau = 3, \lambda_t = 0.1$ and skip optimization as domain discrepancy between background and foreground images is negligible.

## 5.2 Performance Comparisons

We compare TALE with prior SOTA and concurrent works that are capable of performing image-guided composition, including TF-ICON [29], SDEdit [30], Blended Diffusion [1], Paint by Example [49], AnyDoor [3], ControlCom [52], and ObjectStitch [42].

**Qualitative Results.** Qualitative results shown in Fig. 4 highlight the superiority of our method across all domains. First, TALE generates high-quality composited images of which the objects are stylized according to target backgrounds more naturally. Second,

the identity features of input objects are better preserved. Third, the complementing background regions of composited images remain unchanged. Fourth, the objects seamlessly blend into the backgrounds without noticeable artifacts in the transition area. In one hand, although AnyDoor, ControlCom, and ObjectStitch can compose images within their photorealistic training domain, they suffer from poor adaptation to other domains. On the other hand, TF-ICON and Paint by Example can provide certain degree of freedom for composing in different domains yet they fall short in retaining object identities and altering color style. For SDEdit and Blended Diffusion, while the former often causes unwanted changes to the background, the latter solely resorts to text prompt for composing; hence, its results tend to deviate from user's intention.

**Quantitative Results.** We first follow the prior works to perform quantitative comparisons using four metrics: LPIPS$_{bg}$ [55] to assess background preservation, LPIPS$_{fg}$ [55] to measure low-level similarity between foreground image and the edited region, CLIP$_{Image}$ [36] to examine the semantic correspondence between foreground image and the edited region in CLIP embedding space, and CLIP$_{Text}$ [36] to evaluate the semantic alignment between input text prompt and the composited image. However, since these metrics do not assess domain style adaptability and are known for texture and semantic bias [8, 18] in which style information can affect the scores, we only employ them for evaluating composition within the same photorealism domain. As demonstrated in Tab. 1, our method TALE achieves the best performance among training-free approaches, even outperforms several frameworks that are trained on this domain.

For cross-domain comparisons, we adopt the recent evaluation protocol from [18], which can precisely examine domain transferability in terms of style and content similarity. Specifically, we leverage their pre-trained discriminator to predict color style similarity score between the edited patch of composited image and the background. For content similarity, we utilize LDC [44] model to extract edge features of background, foreground, and composited images. These features are more tolerant of style changes and hence can be used to assess content preservation. We then compute content similarity score with a slight modification as

$$\mathcal{S} = (1 + \text{SSIM}_{bg})(1 + \text{SSIM}_{fg})/4, \quad (13)$$

where $\text{SSIM}_{fg}$ denotes SSIM calculated between edge features of

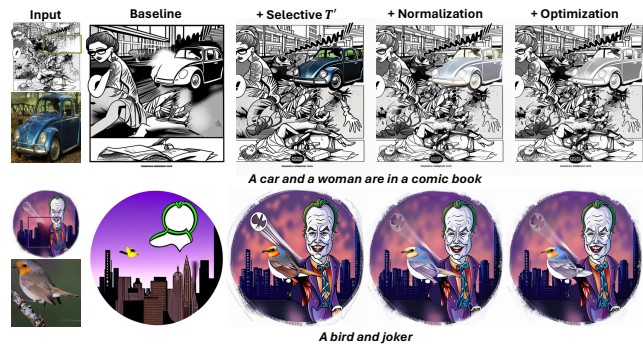

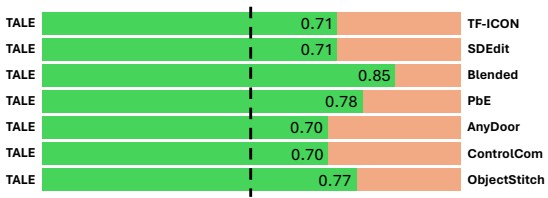

Figure 6: Ablation study: Qualitative evaluation on effectiveness of each component.

Figure 7: User preference of TALE over prior works.

foreground image and edited region of resulting image, and $SSIM_{bg}$ is calculated on the complementing background area. This metric formulation can effectively reflect both background and object identity preservation capabilities. Results presented in Fig. 5 show that we attain the most balanced content-style trade-off across all domains. We can observe that although Anydoor and ControlCom have high content similarity scores, they often fail to alter the object style. In contrast, SDEdit may obtain high style similarity scores yet they struggle to retain content information.

**User Study.** To subjectively evaluate the performance of our TALE compared to other methods, we invite 50 users to participate in a user study. We show each of them 20 to 30 image sets randomly selected from a pool of 310 questions each consists of a background image, a foreground image, and two composited options of which one is from ours and the other is randomly picked from 7 results generated by prior works. Users are required to select the better-composited image based on comprehensive criteria considered foreground content-style balance, background preservation, text alignment, and seamless composition. After collecting user responses, we computed the average preference percentage of our method over others. Fig. 7 shows that TALE is greatly favored by the users.

### 5.3 Ablation Studies

**Component Effectiveness.** We sequentially ablate the key elements of our proposed TALE on the extended dataset with the following configurations: (1) Baseline, in which the composition is generated by a plain denoising process from $T$ to 0 with neither adaptive latent manipulation nor energy-guided optimization. The initial point is composed by incorporating inverted noises at $T' = T$; (2) $T'$ is selectively set; (3) The adaptive normalization is additionally conducted; (4) The energy-guided optimization is

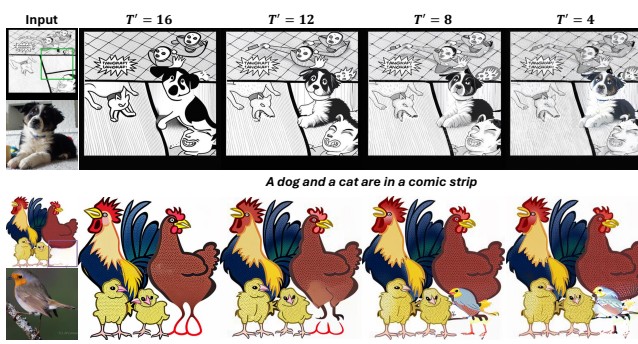

Figure 8: Ablation study: Qualitative evaluation on different selections of $T'$.

**Table 2: Ablation study: Quantitative evaluation on effectiveness of each component.**

| Config | Baseline | + Selective T' | + Normalization | + Optimization |
|---|---|---|---|---|
| Content Similarity ↑ | 0.45 | 0.48 (+ 0.03) | 0.49 (+ 0.01) | **0.50** (+ 0.01) |
| Style Similarity ↑ | 0.40 | 0.50 (+ 0.10) | 0.81 (+ 0.31) | **0.82** (+ 0.01) |

**Table 3: Ablation study: Quantitative evaluation on different selection of $T'$.**

| Config | $T' = 16$ | $T' = 12$ | $T' = 8$ | $T' = 4$ |
|---|---|---|---|---|
| Content Similarity ↑ | 0.47 | 0.48 | 0.50 | **0.51** |
| Style Similarity ↑ | 0.56 | 0.75 | **0.82** | 0.78 |

finally applied. Results shown in Tab. 2 and Fig. 6 indicate that the proper selection of $T'$ can preserve content and style information of inputs while adaptive normalization can enhance the color tone of objects and energy-guided optimization helps further refine the outcomes.

$T'$ **Selection.** Intuitively, the more the denoising progresses, the more information about backgrounds and objects are reconstructed, hence the more effectively they can be composed into final outcomes. To validate this intuition, we experiment with the influence of different choices of $T'$ on the extended dataset. Consistent results are demonstrated in Fig. 8 and Tab. 3. Notably, too large $T'$ leads to content information loss, while too small $T'$ affects domain style adaptation.

## 6 CONCLUSION

We have presented a novel training-free framework dubbed TALE leveraging powerful text-driven diffusion models for high-quality cross-domain image-guided composition. TALE is equipped with two components, namely Adaptive Latent Manipulation and Energy-guided Latent Optimization, that works in synergy to construct and control the composition process, seamlessly incorporating user-provided objects into a specific visual background of different domains. Our experimental results highlight the superiority of our approach over prior and concurrent works, achieving state-of-the-art performance. We hope that our method can inspire future research on similar or relevant topics.

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
