# OpenReview forum: "TALE: Training-free Cross-domain Image Composition via Adaptive Latent Manipulation and Energy-guided Optimization"
_acmmm.org/ACMMM/2024/Conference — MM2024 Poster_

### Official Review · Reviewer_ENg3 · 2024-05-23

**Rating:** 4
**Confidence:** 3

**Summary:**

This paper introduces TALE, a training-free framework using text-driven diffusion models for cross-domain image composition. TALE proposes two important tricks to improve TF-ICON.  First, it employs Adaptive Latent Manipulation to initiate and steer composition. Then, it presents Energy-guided Latent Optimization to refine intermediate latents. Their experiments show that TALE outperforms existing methods, achieving state-of-the-art results in both photorealistic and artistic domains.

**Strengths:**

1. The paper is well-written and easy to follow.
2. Most experiments are solid and the results are fantastic. The ablation studies show how important the three components are.
3. The ideas of the Adaptive Latent Manipulation and Energy-guided Latent Optimization are simple but efficient. I am convinced that these two contributions help to improve the correctness of TF-ICON.

**Limitations:**

1. The contributions are not so novel. The paper shows the importance of Selective 𝑻′, Normalization, and Energy-guided Latent Optimization in Image Composition. The ideas are not so novel and frequently used in diffusion models. I think a fixed $T$' and a fixed Optimization step $N$  may not be suitable for all images. Therefore, I recommend the authors improve the method by discussing how to adaptively choose $T$' and $N$ for each image instance.
2.  It seems that the ablation studies do not show the specific functions of each component. I am a little confused about the Figure 6. The authors say that Normalization is used for addressing significant domain discrepancies such as black-and-white $x_{bg}$ and colorful $x_{fg}$. However, in Figure 6, the result after adding Normalization is still colorful while the result after adding Optimization becomes black-and-white. I hope the authors give some explanation to the case.
3. Some small grammar issues exist in the paper. For example, $c$ in line 294 and $\mathbf{c}$ in line 297 are not consistent.

**Suitability:**

3

---

### Official Review · Reviewer_KQc8 · 2024-05-24

**Rating:** 3
**Confidence:** 4

**Summary:**

This paper presents TALE, a novel training-free framework that leverages the power of text-driven diffusion models to tackle the task of cross-domain image composition. TALE is a training-free method that operates directly on the latent space to provide explicit and effective guidance for the composition process. It is equipped with two mechanisms:
1. Adaptive Latent Manipulation: Formulates noisy latents conducive to initiating and steering the composition process by directly leveraging background and foreground latents at corresponding timesteps.
2. Energy-guided Latent Optimization: Exploits designated energy functions to further optimize intermediate latents to generate desired final results.

**Strengths:**

1. Adequate evaluation: The paper conducts extensive experiments, comparing the proposed method with several state-of-the-art approaches across different domains. Quantitative metrics and qualitative results are provided to support the claims.
2. Applications: Suitable for more domain of images:1. This paper extend the image the baseline benchmark with more non-photorealistic samples and diverse backgrounds, and it outperforms on these non-photorealistic domain compared to existing methods.

**Limitations:**

1. Lack of Novelty: The paper builds upon previous work on image composition, which is similar to TF-ICON , which have also explored the use of a training-free method through the manipulation of attention map in the diffusion models. The main improvement lies in the selection of inversion timesteps to employ
and it involves the AdaIn normalization, which is not the contribution of this work. It's not the first time to use the AdaIN normalization in style transfer [1] [2]. Besides, the energy-guided latent Optimization is quite naive and has none analysis why this is work for this task.
2. Insufficient Evaluation: The paper primarily focuses on presenting the technical details of TALE and its performance compared to prior methods. However, the evaluation could be strengthened by including.
3. Theoretical Approach: the section of energy-guided latent Optimization could benefit from more rigorous mathematical formulations and proofs to strengthen the theoretical arguments.
Chung, Jiwoo, Sangeek Hyun, and Jae-Pil Heo. "Style Injection in Diffusion: A Training- free Approach for Adapting Large-scale Diffusion Models for Style Transfer." arXiv preprint arXiv:2312.09008 (2023) (CVPR 2024).

**Suitability:**

3

---

### Official Review · Reviewer_ZXpS · 2024-05-26

**Rating:** 4
**Confidence:** 3

**Summary:**

This work proposes TALE for image composition. TALE features adaptive latent manipulation and energy-guided optimization in a training-free manner. Experimental results show that TALE outperforms previous methods, especially in cross-domain cases.

**Strengths:**

1. TALE can handle cross-domain image composition, where many previous methods failed.
2. TALE does not need training before composing images.
3. The performance of TALE seems good.

**Limitations:**

### Major issues
1. It would be better if the authors could further clarify the denoising schedule. Does TALE adopt an accelerated denoising process like DDIM? Does $T = 20$ in Line 693 correspond to the full noise at (conventionally) time step 1000?
2. The adaptive latent normalization in Sec. 4.2 operates on latent space. However for Stable Diffusion (since the authors did not specify which pretrained LDM was used in TALE, I suppose it might be SD), latent codes usually have just a few channels (like 3 or 4). In such case, I am not sure if AdaIN is capable of learning rather high-level style information (e.g. painting styles, instead of low-level style like colors).
3. It seems to me that the text prompts do not provide additional information as they are just describing what's in the images. Hence I am not sure if text prompts are really necessary for image composition.

### Minor issues and suggestions
4. In Sec. 4, $\odot$ and $\oplus$ are not clearly defined.
5. In Tab. 1--3 TALE and the baselines achieve scores that are very close for some of the metrics (e.g. LPIPS and content similarity). I am afraid that two digits after the decimal point are not sufficient in such cases and the authors should report more significant figures.

**Suitability:**

2

---

### Official Review · Reviewer_wRkm · 2024-05-29

**Rating:** 4
**Confidence:** 4

**Summary:**

This work introduces a training-free framework, TALE, to incorporate objects into a visual context. TALE is comprised of two stages, Adaptive Latent Manipulation and Energy-guided Latent Optimization. Adaptive latent manipulation starts denoising at $T'$, where $0<T'<T$, and composes the object and the background directly with object mask and without transition area mask. Adaptive latent manipulation also introduces a latent normalization to reduce the significant domain discrepancy exists between object and background. Energy-guided latent optimization introduces CLIP as a pre-trained multimodal projector $\mathscr{P}$ to measure the similarity between predicted image and prompt as extra guidance. Comprehensive experiments show effectiveness of TALE.

**Strengths:**

- The overall paper is easy to read and clear.
- Comprehensive experiments have been done to show effectiveness.
- Method is training free.

**Limitations:**

- The method only showcases single object incorporation cases. Multiple objects case might be interesting to see.
- The prompts guidance are plain description of object and background, which are easy cases without interactions. And the output looks like style-based copy paste. More challenging cases will be interesting to see.

**Suitability:**

3

---

### Meta-Review · Senior_Area_Chairs · 2024-07-09

**Recommendation:** Accept (Poster)
**Confidence:** 5

**Metareview:**

In this paper the author present a novel training-free framework harnessing the power of text-driven diffusion models to tackle a cross-domain image imposition task that aims at seamlessly incorporating user-provided objects into a specific visual context. Reviewers appreciate the comprehensive evaluation and the quality of the results. There were some issues raised and after rebuttal they have been mostly resolved and three of the reviewers give it a borderline accept and the fourth raised it to weak accept.